# Is the information geometry of
# probabilistic population codes learnable?

**John J. Vastola**                                    JOHN_VASTOLA@HMS.HARVARD.EDU
**Zach Cohen**                                            ZCOHEN1@G.HARVARD.EDU
**Jan Drugowitsch**                                  JAN_DRUGOWITSCH@HMS.HARVARD.EDU
*Department of Neurobiology, Harvard Medical School, Boston, MA, USA*

**Editors:** Sophia Sanborn, Christian Shewmake, Simone Azeglio, Arianna Di Bernardo, Nina Miolane

## Abstract

One reason learning the geometry of latent neural manifolds from neural activity data is difficult is that the ground truth is generally not known, which can make manifold learning methods hard to evaluate. Probabilistic population codes (PPCs), a class of biologically plausible and self-consistent models of neural populations that encode parametric probability distributions, may offer a theoretical setting where it is possible to rigorously study manifold learning. It is natural to define the neural manifold of a PPC as the statistical manifold of the encoded distribution, and we derive a mathematical result that the information geometry of the statistical manifold is directly related to measurable covariance matrices. This suggests a simple but rigorously justified decoding strategy based on principal component analysis, which we illustrate using an analytically tractable PPC.

**Keywords:** Probabilistic population codes, information geometry, manifold learning

## 1. Introduction

A common goal in systems neuroscience is to understand the information encoded in the heterogeneous responses of large populations of neurons. A key observation is that the dimensionality of neural responses—determined by the size and stochasticity of a given population encoding a variable of interest—is much larger than a population's *effective* dimensionality, or the number of dimensions needed to describe a majority of response variation (Gao and Ganguli, 2015; Nieh et al., 2021; Chaudhuri et al., 2019; Gardner et al., 2022). This lower-dimensional effective space is commonly called a *(latent) neural manifold*, owing to the fact that the bases of the effective space may not be Euclidean.

The geometry of a neural manifold offers a lens into the computations performed by the corresponding population of neurons (Chung and Abbott, 2021). Learning this geometry from neural activity alone is a generally difficult and ill-defined inverse problem, and as such, requires the use of strong assumptions. For example, it is often assumed that task-relevant stimulus variables can be linearly decoded from population activity (Ma et al., 2006). Although a variety of manifold learning methods have been empirically successful at extracting useful geometric insight (Low et al., 2018; Tenenbaum et al., 2000; Gardner et al., 2022), it is generally challenging to establish a principled mathematical connection between the information encoded in a neural population's activity and its associated latent geometry. In this work, we demonstrate such a connection in the well-studied theoretical setting of probabilistic population codes.

Probabilistic population codes (PPCs) are a proposed coding mechanism whereby populations of neurons encode the parameters of probability distributions (Ma et al., 2006). PPCs model spike generation in a biologically plausible way (Figure 1a), are mathematically self-consistent, and are flexible, allowing a variety of tuning curves and neural correlation structures. These properties have made them attractive for modeling real-world neural recordings (Beck et al., 2008; Hou et al., 2019). For understanding neural geometry, a particular benefit of studying PPCs is that there is an intuitive candidate for the latent neural manifold: the *statistical manifold* of the represented distribution (Figure 1b).

The statistical manifold hails from the field of information geometry, which has found increasing application in the fields of machine learning (Martens, 2014; Zhang et al., 2019; Karakida and Osawa, 2020; Oizumi et al., 2016) and computational neuroscience (Kreutzer et al., 2022). A statistical manifold is a manifold whose points are parameterizations of a particular probability distribution (Amari, 1998, 2002, 2016); for example, one point on a normal distribution's statistical manifold corresponds to a particular mean and standard deviation. Any parametric distribution has a corresponding statistical manifold whose coordinates are distribution parameters, and whose metric is the Fisher information matrix of that distribution. In this paper, we confirm analytically that the statistical manifold is indeed an appropriate assignment for the latent neural manifold of a PPC, and that the manifold dimensionality, as well as its metric, can in principle be learned by measuring tuning curves and computing covariance matrices from neural data (Figure 1c).

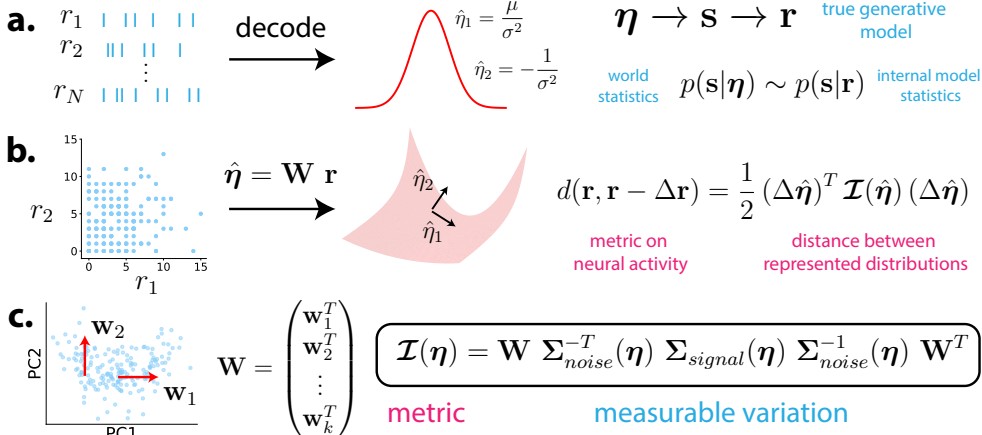

Figure 1: PPCs and the proposed geometry recovery strategy. **a**. The neural activity of PPCs encodes a parametric probability distribution $p(\boldsymbol{s}|\boldsymbol{r})$, which is assumed to be similar to the true distribution $p(\boldsymbol{s}|\boldsymbol{\eta})$ of some observed stimulus $\boldsymbol{s}$. The stimulus and $\boldsymbol{r}$ relate through the generative model $\boldsymbol{\eta} \to \boldsymbol{s} \to \boldsymbol{r}$. **b**. In linear PPCs, distribution parameters $\hat{\boldsymbol{\eta}}$ can be linearly decoded from neural activity $\boldsymbol{r}$, and the space of decoded parameters can be viewed as a statistical manifold whose metric comes from the distribution's Fisher information matrix. **c**. In this work, we propose a principal-component-analysis-like strategy for estimating the metric by measuring neural activity covariance matrices.

## 2. Probabilistic population codes

Probabilistic population codes (PPCs) represent parametric distributions from a broad class of probability distributions known as the *exponential family*. A general (canonical form) exponential family likelihood has the form

$$p(\boldsymbol{s}|\boldsymbol{\eta}) = h \exp\left\{\boldsymbol{\eta}^T \boldsymbol{T}(\boldsymbol{s}) - A(\boldsymbol{\eta})\right\} \tag{1}$$

where $\boldsymbol{s} \in \mathbb{R}^S$ is the stimulus vector, $\boldsymbol{\eta} \in \mathbb{R}^k$ is the vector of natural parameters, $\boldsymbol{T}(\boldsymbol{s}) \in \mathbb{R}^k$ is the vector of sufficient statistics, $A(\boldsymbol{\eta})$ is the log-partition function, and $h$ is the base measure. For simplicity, we will assume that the base measure is constant, although this assumption can be relaxed. (For an example, see Appendix A.)

Linear PPCs are assumed to represent an exponential family stimulus likelihood in the sense that they model the probability of the stimulus taking a certain value, given neural activity $\boldsymbol{r} \in \mathbb{R}^N$ in a population of $N \gg k$ neurons, as (Ma et al., 2006)

$$p(\boldsymbol{s}|\boldsymbol{r}) = h \exp\left\{(\boldsymbol{W}\boldsymbol{r})^T \boldsymbol{T}(\boldsymbol{s}) - A(\boldsymbol{W}\boldsymbol{r})\right\} \tag{2}$$

where $\boldsymbol{W} \in \mathbb{R}^{k \times N}$ is the natural parameter readout matrix. The 'linear' designation refers to the fact that the natural parameters, which determine the specific distribution being represented, can be linearly read out via $\hat{\boldsymbol{\eta}} := \boldsymbol{W}\boldsymbol{r}$. (For an example, see Appendix B.)

In order for $p(\boldsymbol{s}|\boldsymbol{r})$ to take a particular form, there must be fairly strong constraints on neural responses to stimuli, i.e. $p(\boldsymbol{r}|\boldsymbol{s})$. Two such necessary (but not sufficient) constraints, which can be derived in various ways (see Appendix C), are

$$0 = \boldsymbol{J}_{\boldsymbol{T}}^T(\boldsymbol{s})\ \boldsymbol{W}\ \boldsymbol{f}(\boldsymbol{s}) \tag{3}$$
$$\boldsymbol{J}_{\boldsymbol{f}}^T(\boldsymbol{s}) = \boldsymbol{J}_{\boldsymbol{T}}^T(\boldsymbol{s})\ \boldsymbol{W}\ \boldsymbol{\Sigma}(\boldsymbol{s})$$

where $\boldsymbol{f}(\boldsymbol{s}) := \langle \boldsymbol{r} \rangle_{p(\boldsymbol{r}|\boldsymbol{s})} \in \mathbb{R}^N$ are the tuning curves, $\boldsymbol{\Sigma}(\boldsymbol{s}) := \mathrm{Cov}(\boldsymbol{r}, \boldsymbol{r})_{p(\boldsymbol{r}|\boldsymbol{s})}$ is the $N \times N$ fixed-stimulus covariance matrix, $\boldsymbol{J}_{\boldsymbol{T}}(\boldsymbol{s})$ is the $k \times S$ Jacobian of $\boldsymbol{T}(\boldsymbol{s})$, and $\boldsymbol{J}_{\boldsymbol{f}}(\boldsymbol{s})$ is the $N \times S$ Jacobian of $\boldsymbol{f}(\boldsymbol{s})$. For example, for a population of independent Poisson neurons, the first condition says that the tuning curves sum to a stimulus-independent value.

Although the form of the encoded distribution places strong constraints on neural activity, it is not clear from prior work which neural activity measurements make it possible, at least in principle, to determine the distribution a population is representing. For example, is it sufficient to measure tuning curves and covariance matrices, as is commonly done?

## 3. The information geometry of probabilistic population codes

Parametric probability distributions, like those in the exponential family, can be associated with Riemannian manifolds called *statistical manifolds*. The metric on such manifolds is the Fisher information matrix $\boldsymbol{\mathcal{I}}(\boldsymbol{\eta})$, whose components are

$$[\boldsymbol{\mathcal{I}}(\boldsymbol{\eta})]_{i,j} := -\mathbb{E}\left[\frac{\partial^2}{\partial \eta_i \partial \eta_j} \log p(\boldsymbol{s}|\boldsymbol{\eta})\right]_{p(\boldsymbol{s}|\boldsymbol{\eta})} = \frac{\partial^2 A(\boldsymbol{\eta})}{\partial \eta_i \partial \eta_j} \tag{4}$$

for exponential families. In other words, $\boldsymbol{\mathcal{I}}(\boldsymbol{\eta}) = \boldsymbol{H}_A(\boldsymbol{\eta})$, the $k \times k$ Hessian of the log-partition function $A$.

Is the information geometry of the distribution represented by a PPC related to a more basic geometry we could associate with neural activity? Intuitively, since $\boldsymbol{r}$ represents a probability distribution over the stimulus $\boldsymbol{s}$, two neural activity vectors should be considered somewhat different if the represented distribution is somewhat different.

This motivates defining a geometry whose metric, as in the case of statistical manifolds, locally captures the discrepancy between represented probability distributions. That is, we would like a metric that is locally equal to the Kullback-Leibler (KL) divergence between $p(\boldsymbol{s}|\boldsymbol{r})$ and $p(\boldsymbol{s}|\boldsymbol{r} - \Delta\boldsymbol{r})$, for $\Delta\boldsymbol{r}$ sufficiently small compared to typical values of $\boldsymbol{r}$:

$$d(\boldsymbol{r}, \boldsymbol{r} - \Delta\boldsymbol{r}) := D_{KL}(\ p(\boldsymbol{s}|\boldsymbol{r})\ \|\ p(\boldsymbol{s}|\boldsymbol{r} - \Delta\boldsymbol{r})\ ) = \langle\ \log p(\boldsymbol{s}|\boldsymbol{r}) - \log p(\boldsymbol{s}|\boldsymbol{r} - \Delta\boldsymbol{r})\ \rangle_{p(\boldsymbol{s}|\boldsymbol{r})}\ . \quad (5)$$

If $\Delta\boldsymbol{r}$ is small,

$$\log p(\boldsymbol{s}|\boldsymbol{r} - \Delta\boldsymbol{r}) \approx \log p(\boldsymbol{s}|\boldsymbol{r}) - (\Delta\boldsymbol{r})^T \boldsymbol{W}^T [\ \boldsymbol{T}(\boldsymbol{s}) - \nabla_{\boldsymbol{\eta}} A(\boldsymbol{\eta})\ ] - \frac{1}{2}(\Delta\boldsymbol{r})^T \boldsymbol{W}^T \boldsymbol{H}_A(\boldsymbol{\eta}) \boldsymbol{W}(\Delta\boldsymbol{r})\ ,$$

so

$$\begin{aligned} D_{KL} &= (\Delta\boldsymbol{r})^T \boldsymbol{W}^T [\ \langle\boldsymbol{T}(\boldsymbol{s})\rangle - \nabla_{\boldsymbol{\eta}} A(\boldsymbol{\eta})\ ] + \frac{1}{2}(\Delta\boldsymbol{r})^T \boldsymbol{W}^T \boldsymbol{H}_A(\boldsymbol{\eta}) \boldsymbol{W}(\Delta\boldsymbol{r}) \qquad (6)\\ &= \frac{1}{2}\ (\Delta\boldsymbol{r})^T \boldsymbol{W}^T \boldsymbol{H}_A(\boldsymbol{\eta}) \boldsymbol{W}(\Delta\boldsymbol{r}) \\ &= \frac{1}{2}\ (\Delta\hat{\boldsymbol{\eta}})^T \boldsymbol{H}_A(\boldsymbol{\eta})(\Delta\hat{\boldsymbol{\eta}}) \end{aligned}$$

where we have used the fact that $\langle\boldsymbol{T}(\boldsymbol{s})\rangle_{p(\boldsymbol{s}|\boldsymbol{\eta})} = \nabla_{\boldsymbol{\eta}} A(\boldsymbol{\eta})$. From the above, we can see that the natural metric on neural activity exactly corresponds to a projected version of the natural metric on the corresponding statistical manifold. This makes it natural to *define* the neural manifold of a PPC as the corresponding statistical manifold. But while this assignment is theoretically interesting, does this geometry correspond to anything measurable? In what follows, we pursue the question of learning $\boldsymbol{W}$ and $\boldsymbol{H}_A(\boldsymbol{\eta})$ from neural data.

## 4. Neural correlations reflect information geometry

Fluctuations in neural activity occur for two different reasons: firstly, because neural activity varies even when the stimulus is held fixed; and secondly, because the external stimulus varies. The first kind of fluctuations are called *noise correlations*, while the second are called *signal correlations*. Mathematically, the law of total covariance allows us to write the total covariance matrix as a sum of the two types of variation[1]:

$$\begin{aligned} \mathrm{Cov}(\boldsymbol{r}, \boldsymbol{r})_{p(\boldsymbol{r}|\boldsymbol{\eta})} &= \boldsymbol{\Sigma}_{noise}(\boldsymbol{\eta}) + \boldsymbol{\Sigma}_{signal}(\boldsymbol{\eta}) \qquad (7)\\ &:= \langle\ \boldsymbol{\Sigma}(\boldsymbol{s})\ \rangle_{p(\boldsymbol{s}|\boldsymbol{\eta})} + \mathrm{Cov}(\ \boldsymbol{f}(\boldsymbol{s}), \boldsymbol{f}(\boldsymbol{s})\ )_{p(\boldsymbol{s}|\boldsymbol{\eta})}\ . \end{aligned}$$

Naively, we might expect that noise correlations are not particularly informative about latent dynamics on the neural manifold (i.e. natural parameter changes), since the corresponding fluctuations happen even if the distribution represented by neural activity remains

---

1. One should be careful to note that, unlike how e.g. signal covariance is usually defined, here our matrices are conditional on $\boldsymbol{\eta}$.

the same. Conversely, signal correlations *should* be informative about latent dynamics, since changes in the stimulus should yield changes in the represented distribution, and hence neural activity changes.

However, these two types of variation are not completely independent. Because neurons that tend to vary together when the stimulus is fixed also tend to have their mean activities change in a correlated way when the stimulus changes, signal correlations are to some extent 'contaminated' by noise correlations. If this contamination can be 'undone', the signal correlation matrix might be expected to reflect the information geometry of the latent statistical manifold.

In the case of linear PPCs, this intuition can be made precise. We will show that

$$\boldsymbol{\Sigma}_{noise}^{-T}(\boldsymbol{\eta}) \; \boldsymbol{\Sigma}_{signal}(\boldsymbol{\eta}) \; \boldsymbol{\Sigma}_{noise}^{-1}(\boldsymbol{\eta}) \approx \boldsymbol{W}^T \boldsymbol{H}_A(\boldsymbol{\eta}) \boldsymbol{W} = \boldsymbol{W}^T \boldsymbol{\mathcal{I}}(\boldsymbol{\eta}) \boldsymbol{W} \; . \tag{8}$$

The left-hand side is the signal covariance matrix 'adjusted' for the effect of noise correlations, while the right-hand side is the natural metric on the latent statistical manifold projected into the space of neural activity (Equation (6)). This equation represents our link between information geometry and measurable quantities.

To show this, we will need to compute the noise and signal correlation matrices for an arbitrary linear PPC. A useful fact is that, for an arbitrary vector $\boldsymbol{v} \in \mathbb{R}^{k \times 1}$,

$$\langle e^{\boldsymbol{v}^T \boldsymbol{T}(\boldsymbol{s})} \rangle_{p(\boldsymbol{s}|\boldsymbol{\eta})} = \int h \exp\left\{ (\boldsymbol{\eta} + \boldsymbol{v})^T \boldsymbol{T}(\boldsymbol{s}) - A(\boldsymbol{\eta}) \right\} \; d\boldsymbol{s} = \exp\left\{ A(\boldsymbol{\eta} + \boldsymbol{v}) - A(\boldsymbol{\eta}) \right\} \; . \tag{9}$$

The formal computation of the covariance matrices will be tractable if it can be reduced to computing integrals of the above form. Fortunately, using a somewhat technical generating-function-based argument, we can derive results that facilitate this strategy (see Appendix C). The tuning curves and fixed-stimulus covariance matrix can be written as infinite series

$$\boldsymbol{f}(\boldsymbol{s}) := \langle \boldsymbol{r} \rangle_{p(\boldsymbol{r}|\boldsymbol{s})} = \sum_{\boldsymbol{n} \in \mathbb{N}^N} \boldsymbol{n} \; c_{\boldsymbol{n}} \; e^{\boldsymbol{n}^T \boldsymbol{W}^T \boldsymbol{T}(\boldsymbol{s})} \tag{10}$$

$$\boldsymbol{\Sigma}(\boldsymbol{s}) := \text{Cov}(\boldsymbol{r}, \boldsymbol{r})_{p(\boldsymbol{r}|\boldsymbol{s})} = \sum_{\boldsymbol{n} \in \mathbb{N}^N} \boldsymbol{n} \; \boldsymbol{n}^T \; c_{\boldsymbol{n}} \; e^{\boldsymbol{n}^T \boldsymbol{W}^T \boldsymbol{T}(\boldsymbol{s})}$$

for some coefficients $c_{\boldsymbol{n}}$. Using Equations (9) and (10), we can for example compute that

$$\langle \boldsymbol{f}(\boldsymbol{s}) \rangle_{p(\boldsymbol{s}|\boldsymbol{\eta})} = \sum_{\boldsymbol{n} \in \mathbb{N}^N} \boldsymbol{n} \; c_{\boldsymbol{n}} \; \langle e^{\boldsymbol{n}^T \boldsymbol{W}^T \boldsymbol{T}(\boldsymbol{s})} \rangle_{p(\boldsymbol{s}|\boldsymbol{\eta})} \tag{11}$$

$$= \sum_{\boldsymbol{n} \in \mathbb{N}^N} \boldsymbol{n} \; c_{\boldsymbol{n}} \; e^{A(\boldsymbol{\eta} + \boldsymbol{W}\boldsymbol{n}) - A(\boldsymbol{\eta})} \; .$$

Similarly, we can compute that the noise covariance matrix is

$$\boldsymbol{\Sigma}_{noise}(\boldsymbol{\eta}) = \sum_{\boldsymbol{n} \in \mathbb{N}^N} \boldsymbol{n} \; \boldsymbol{n}^T \; c_{\boldsymbol{n}} \; \langle e^{\boldsymbol{n}^T \boldsymbol{W}^T \boldsymbol{T}(\boldsymbol{s})} \rangle_{p(\boldsymbol{s}|\boldsymbol{\eta})} \tag{12}$$

$$= \sum_{\boldsymbol{n} \in \mathbb{N}^N} \boldsymbol{n} \; \boldsymbol{n}^T \; c_{\boldsymbol{n}} \; e^{A(\boldsymbol{\eta} + \boldsymbol{W}\boldsymbol{n}) - A(\boldsymbol{\eta})}$$

and the signal covariance matrix is

$$\boldsymbol{\Sigma}_{signal}(\boldsymbol{\eta}) = \sum_{\boldsymbol{n},\boldsymbol{m}} \boldsymbol{n}\boldsymbol{m}^T \ c_{\boldsymbol{n}}c_{\boldsymbol{m}} \ \left\{ \langle e^{(\boldsymbol{n}+\boldsymbol{m})^T \boldsymbol{W}^T \boldsymbol{T}(\boldsymbol{s})} \rangle - \langle e^{\boldsymbol{n}^T \boldsymbol{W}^T \boldsymbol{T}(\boldsymbol{s})} \rangle \langle e^{\boldsymbol{m}^T \boldsymbol{W}^T \boldsymbol{T}(\boldsymbol{s})} \rangle \right\} \tag{13}$$

$$= \sum_{\boldsymbol{n},\boldsymbol{m}} \boldsymbol{n}\boldsymbol{m}^T \ c_{\boldsymbol{n}}c_{\boldsymbol{m}} \ \left\{ e^{A(\boldsymbol{\eta}+\boldsymbol{W}(\boldsymbol{n}+\boldsymbol{m})) - A(\boldsymbol{\eta})} - e^{A(\boldsymbol{\eta}+\boldsymbol{W}\boldsymbol{n}) - A(\boldsymbol{\eta}) + A(\boldsymbol{\eta}+\boldsymbol{W}\boldsymbol{m}) - A(\boldsymbol{\eta})} \right\} \ .$$

At this point, we must make an approximation. We will assume that the components of the readout matrix $\boldsymbol{W}$ are small; intuitively, this means that each of the many neurons in the population typically contributes somewhat to the parameter estimate $\hat{\boldsymbol{\eta}} = \boldsymbol{W}\boldsymbol{r}$. Assuming that $\boldsymbol{W}$ is small allows us to Taylor expand the log-partition function as e.g.

$$A(\boldsymbol{\eta} + \boldsymbol{W}\boldsymbol{n}) - A(\boldsymbol{\eta}) \approx (\nabla_{\boldsymbol{\eta}} A)^T \boldsymbol{W}\boldsymbol{n} + \frac{1}{2} \ (\boldsymbol{W}\boldsymbol{n})^T \boldsymbol{H}_A(\boldsymbol{\eta})(\boldsymbol{W}\boldsymbol{n}) \ . \tag{14}$$

This means the bracketed expression in $\boldsymbol{\Sigma}_{signal}$ is approximately

$$e^{(\nabla A)^T \boldsymbol{W}(\boldsymbol{n}+\boldsymbol{m})} \left\{ e^{\frac{1}{2} \ [\boldsymbol{W}(\boldsymbol{n}+\boldsymbol{m})]^T \boldsymbol{H}_A [\boldsymbol{W}(\boldsymbol{n}+\boldsymbol{m})]} - e^{\frac{1}{2} \ (\boldsymbol{W}\boldsymbol{n})^T \boldsymbol{H}_A (\boldsymbol{W}\boldsymbol{n}) + \frac{1}{2} \ (\boldsymbol{W}\boldsymbol{m})^T \boldsymbol{H}_A (\boldsymbol{W}\boldsymbol{m})} \right\}$$

$$\approx \ \frac{1}{2} \left\{ [\boldsymbol{W}(\boldsymbol{n}+\boldsymbol{m})]^T \boldsymbol{H}_A [\boldsymbol{W}(\boldsymbol{n}+\boldsymbol{m})] - (\boldsymbol{W}\boldsymbol{n})^T \boldsymbol{H}_A (\boldsymbol{W}\boldsymbol{n}) - (\boldsymbol{W}\boldsymbol{m})^T \boldsymbol{H}_A (\boldsymbol{W}\boldsymbol{m}) \right\}$$

$$= (\boldsymbol{W}\boldsymbol{n})^T \boldsymbol{H}_A (\boldsymbol{W}\boldsymbol{m})$$

to second order in $\boldsymbol{W}$. Then $\boldsymbol{\Sigma}_{signal}$ can be written as

$$\boldsymbol{\Sigma}_{signal}(\boldsymbol{\eta}) \approx \sum_{\boldsymbol{n},\boldsymbol{m}} c_{\boldsymbol{n}}c_{\boldsymbol{m}} \ \boldsymbol{n}(\boldsymbol{W}\boldsymbol{n})^T \boldsymbol{H}_A(\boldsymbol{\eta})(\boldsymbol{W}\boldsymbol{m})\boldsymbol{m}^T \tag{15}$$

$$= \sum_{\boldsymbol{n},\boldsymbol{m}} c_{\boldsymbol{n}}c_{\boldsymbol{m}} \ \boldsymbol{n}\boldsymbol{n}^T \left( \boldsymbol{W}^T \boldsymbol{H}_A(\boldsymbol{\eta})\boldsymbol{W} \right) \boldsymbol{m}\boldsymbol{m}^T$$

$$= \left( \sum_{\boldsymbol{n}} c_{\boldsymbol{n}} \ \boldsymbol{n}\boldsymbol{n}^T \right) \left( \boldsymbol{W}^T \boldsymbol{H}_A(\boldsymbol{\eta})\boldsymbol{W} \right) \left( \sum_{\boldsymbol{m}} c_{\boldsymbol{m}} \ \boldsymbol{m}\boldsymbol{m}^T \right) \ .$$

Assuming $\boldsymbol{W}$ is small also means

$$\boldsymbol{\Sigma}_{noise}(\boldsymbol{\eta}) \approx \sum_{\boldsymbol{n} \in \mathbb{N}^N} \boldsymbol{n} \ \boldsymbol{n}^T \ c_{\boldsymbol{n}} \ . \tag{16}$$

To second order in $\boldsymbol{W}$, we then have that

$$\boldsymbol{\Sigma}_{signal}(\boldsymbol{\eta}) \approx \boldsymbol{\Sigma}_{noise}(\boldsymbol{\eta})^T \ \boldsymbol{W}^T \boldsymbol{H}_A(\boldsymbol{\eta})\boldsymbol{W} \ \boldsymbol{\Sigma}_{noise}(\boldsymbol{\eta}) \ . \tag{17}$$

The $N \times N$ noise correlation matrix is by definition symmetric and positive-semidefinite. But in practice it is positive-definite, and hence invertible, so we can invert $\boldsymbol{\Sigma}_{noise}(\boldsymbol{\eta})$ to obtain Equation (8), the desired result.

## 5. Learning neural manifold geometry from neural activity samples

Equation (8) provides a method for learning neural manifold geometry—and hence the represented distribution—using only neural activity samples. We need only assume that the distribution represented by a population of neurons is similar to the ground truth distribution (although not necessarily in form or parameterization).

This method requires two things. First, we must measure the noise and signal covariance matrices, which together allow us to measure the natural metric on neural activity space via Equation (8). Then, to separately identify $\boldsymbol{W}$ and $\boldsymbol{H}_A(\boldsymbol{\eta})$, we must exploit a degeneracy in the definition of exponential family distributions.

### 5.1. The readout matrix is only defined up to an invertible matrix

The degeneracy is that, since $p(\boldsymbol{s}|\boldsymbol{\eta})$ only depends on the dot product of $\boldsymbol{\eta}$ and $\boldsymbol{T}(\boldsymbol{s})$, neither is uniquely defined. In particular, if $\boldsymbol{R}$ is any $k \times k$ invertible linear transformation, and we define $\tilde{\boldsymbol{\eta}} := \boldsymbol{R}\boldsymbol{\eta}$ and $\tilde{\boldsymbol{T}}(\boldsymbol{s}) := \boldsymbol{R}^{-T}\boldsymbol{T}(\boldsymbol{s})$, then

$$\tilde{\boldsymbol{\eta}}^T\tilde{\boldsymbol{T}}(\boldsymbol{s}) = \boldsymbol{\eta}^T\boldsymbol{R}^T\boldsymbol{R}^{-T}\boldsymbol{T}(\boldsymbol{s}) = \boldsymbol{\eta}^T\boldsymbol{T}(\boldsymbol{s}) \ , \tag{18}$$

so the probability distribution (Equation (1)) remains unchanged. The geometry of the statistical manifold is also invariant to such transformations, since

$$(\Delta\tilde{\boldsymbol{\eta}})^T\boldsymbol{H}_A(\tilde{\boldsymbol{\eta}})(\Delta\tilde{\boldsymbol{\eta}}) = (\Delta\boldsymbol{\eta})^T\boldsymbol{R}^T\boldsymbol{R}^{-T}\boldsymbol{H}_A(\boldsymbol{\eta})\boldsymbol{R}^{-1}\boldsymbol{R}(\Delta\boldsymbol{\eta}) = (\Delta\boldsymbol{\eta})^T\boldsymbol{H}_A(\boldsymbol{\eta})(\Delta\boldsymbol{\eta}) \ . \tag{19}$$

For PPCs, this means that $\boldsymbol{W}$ is only identifiable up to an invertible linear transformation.

### 5.2. The readout matrix can be obtained via principal component analysis

This degeneracy in the definition of $\boldsymbol{W}$ can be exploited in the following way. Suppose we have measured an expression of the form $\boldsymbol{W}^T\boldsymbol{M}\boldsymbol{W}$, where $\boldsymbol{M}$ is some real symmetric positive-definite $k \times k$ matrix. This product has a compact singular value decomposition $\boldsymbol{U}^T\boldsymbol{D}\boldsymbol{V}$, where $\boldsymbol{D}$ is diagonal and $\boldsymbol{U}$ and $\boldsymbol{V}$ are $k \times N$ semi-orthogonal matrices (i.e. $\boldsymbol{V}\boldsymbol{V}^T = \boldsymbol{I}_k$). It can be shown that $\boldsymbol{W}$ can always be chosen to be $\boldsymbol{V}$ (see Appendix D).

Which $k \times k$ matrix $\boldsymbol{M}$ should we choose? One appealing choice is the Hessian $\boldsymbol{H}_A(\boldsymbol{\eta})$ averaged over different values of the latent parameter $\boldsymbol{\eta}$. This is similar to doing principal component analysis (PCA) on neural data, since it means considering the truncated eigendecomposition of the average (noise-correlation-adjusted) signal covariance matrix:

$$\langle\ \boldsymbol{\Sigma}_{noise}^{-T}(\boldsymbol{\eta})\ \boldsymbol{\Sigma}_{signal}(\boldsymbol{\eta})\ \boldsymbol{\Sigma}_{noise}^{-1}(\boldsymbol{\eta})\ \rangle_{p(\boldsymbol{\eta})} = \boldsymbol{W}^T\langle\boldsymbol{H}_A(\boldsymbol{\eta})\rangle_{p(\boldsymbol{\eta})}\boldsymbol{W} = \boldsymbol{U}^T\boldsymbol{D}\boldsymbol{V} \ . \tag{20}$$

Such a choice means picking the coordinate system of the latent space so that the statistical manifold is Euclidean on average. This can always be done, since it is always possible to do the corresponding eigendecomposition, but is not crucial for our proposed method to work.

### 5.3. Covariance measurements and the readout matrix determine the metric

Since $\boldsymbol{W}$ is assumed to be semi-orthogonal, i.e. $\boldsymbol{W}\boldsymbol{W}^T = \boldsymbol{I}_k$, it is now trivial to read out the neural manifold metric from the measured covariance matrices, since

$$\boldsymbol{W}\ \boldsymbol{\Sigma}_{noise}^{-T}(\boldsymbol{\eta})\ \boldsymbol{\Sigma}_{signal}(\boldsymbol{\eta})\ \boldsymbol{\Sigma}_{noise}^{-1}(\boldsymbol{\eta})\ \boldsymbol{W}^T = \boldsymbol{W}\boldsymbol{W}^T\boldsymbol{H}_A(\boldsymbol{\eta})\boldsymbol{W}\boldsymbol{W}^T = \boldsymbol{H}_A(\boldsymbol{\eta}) \ . \tag{21}$$

Algorithm 1 summarizes the above steps for learning $\boldsymbol{W}$ and $\boldsymbol{H}_A(\boldsymbol{\eta})$. As an additional technical detail, since $\boldsymbol{\Sigma}_{noise}$ will not contribute any $\boldsymbol{\eta}$-dependence to the signal covariance matrix assuming $\boldsymbol{W}$ is small, to measure it we might as well average over all values of $\boldsymbol{\eta}$ to use as much data as possible:

$$\boldsymbol{\Sigma}_{noise} \approx \langle\ \boldsymbol{\Sigma}_{noise}(\boldsymbol{\eta})\ \rangle_{p(\boldsymbol{\eta})} = \langle\ \boldsymbol{\Sigma}(\boldsymbol{s})\ \rangle_{p(\boldsymbol{s})} = \langle\ (\ \boldsymbol{r} - \boldsymbol{f}(\boldsymbol{s})\ )(\ \boldsymbol{r} - \boldsymbol{f}(\boldsymbol{s})\ )^T\ \rangle_{p(\boldsymbol{r},\boldsymbol{s})}\ . \tag{22}$$

---

**Algorithm 1:** Learning PPC neural manifold metric

---

Given a large collection of $\{\boldsymbol{\eta}_i, \boldsymbol{s}_i, \boldsymbol{r}_i\}$:

1. Measure tuning curves $\boldsymbol{f}(\boldsymbol{s}) := \langle\boldsymbol{r}\rangle_{p(\boldsymbol{r}|\boldsymbol{s})}$.

2. Compute $\boldsymbol{\Sigma}_{noise} \approx \langle\ (\ \boldsymbol{r} - \boldsymbol{f}(\boldsymbol{s})\ )(\ \boldsymbol{r} - \boldsymbol{f}(\boldsymbol{s})\ )^T\ \rangle_{p(\boldsymbol{r},\boldsymbol{s})}$.

3. Compute $\boldsymbol{\Sigma}_{noise}^{-1}$ (where a regularized pseudoinverse is used if necessary).

4. Compute $\boldsymbol{\Sigma}_{signal}(\boldsymbol{\eta}) := \mathrm{Cov}(\ \boldsymbol{f}(\boldsymbol{s}), \boldsymbol{f}(\boldsymbol{s})\ )_{p(\boldsymbol{s}|\boldsymbol{\eta})}$ and $\langle\boldsymbol{\Sigma}_{signal}(\boldsymbol{\eta})\rangle_{p(\boldsymbol{\eta})}$.

5. Diagonalize $\boldsymbol{\Sigma}_{noise}^{-T}\langle\boldsymbol{\Sigma}_{signal}(\boldsymbol{\eta})\rangle_{p(\boldsymbol{\eta})}\boldsymbol{\Sigma}_{noise}^{-1}$ to obtain a decomposition $\boldsymbol{Q}^T\boldsymbol{D}\boldsymbol{Q}$ where $\boldsymbol{Q}$ is orthogonal and $\boldsymbol{D}$ is diagonal. Using a standard elbow-like method, choose to retain the top $k$ components, and obtain a reduced weight matrix $\boldsymbol{O}$ of size $k \times N$. Define $\boldsymbol{W} := \boldsymbol{O}$.

6. For each $\boldsymbol{\eta}$, learn the metric by computing $\boldsymbol{H}_A(\boldsymbol{\eta}) = \boldsymbol{W}\boldsymbol{\Sigma}_{noise}^{-T}\ \boldsymbol{\Sigma}_{signal}(\boldsymbol{\eta})\ \boldsymbol{\Sigma}_{noise}^{-1}\boldsymbol{W}^T$.

---

## 6. Experiments

In this section, we illustrate the metric-learning algorithm using a specific PPC: the minimal correlation model, which is motivated and analytically studied in Appendix E. It has a more complicated neural correlation structure than a population of independent Poisson neurons, but remains analytically tractable, so one can get exact expressions for e.g. the noise correlation matrix, and verify that PPC constraints (e.g. Equation (3)) are satisfied.

We assume that a population of $N = 200$ neurons has statistics described by the minimal correlation model, and encodes a normal distribution with some mean $\mu$ and variance $\sigma^2$ (Figure 2a). One important parameter of the minimal correlation model is $d$, which describes how off-diagonal the structure of the noise correlation matrix is (see Appendix E for more intuition). Define the excess noise correlation matrix

$$\tilde{\boldsymbol{\Sigma}}_{noise}(\boldsymbol{\eta}) := \boldsymbol{\Sigma}_{noise}(\boldsymbol{\eta}) - \mathrm{diag}\left(\langle\boldsymbol{f}(\boldsymbol{s})\rangle_{p(\boldsymbol{s}|\boldsymbol{\eta})}\right)\ , \tag{23}$$

which describes all noise correlation structure not present in a population of independent Poisson neurons of the same size. Analytic expressions from Appendix E can be used to visualize this matrix, along with the signal correlation matrix and two other relevant matrices, to show that Equation (8) is approximately valid (Figure 2b) in a parameter regime for which the components of $\boldsymbol{W}$ are small.

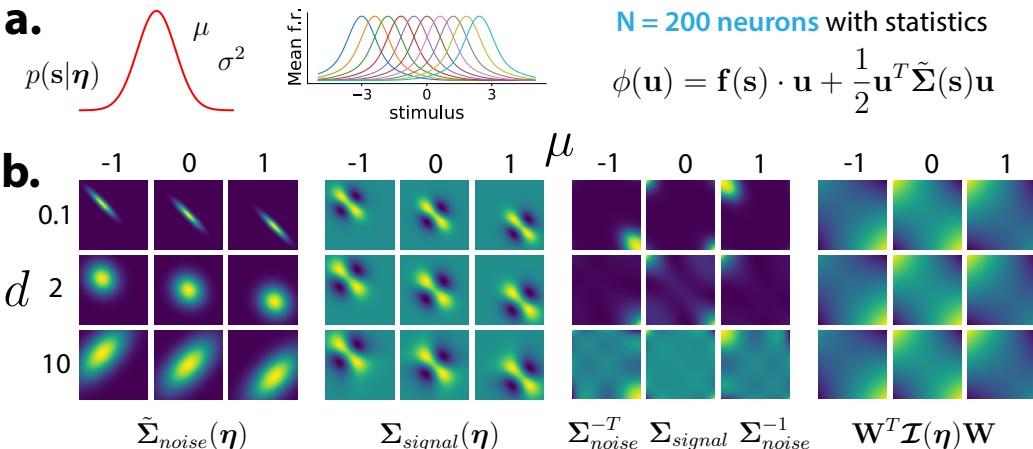

Figure 2: Verifying Equation (8) for the minimal correlation model. **a**. A population of correlated neurons with bump-like tuning curves encodes a normal distribution. **b**. Equation (8) approximately holds for various values of $\mu$ and $d$.

Although the best case scenario for metric recovery is depicted in Figure 3, we found that our algorithm gave inconsistent results. Determining the number of sufficient statistics was robust to e.g. correlation structure, but the ability to obtain a quantitatively correct metric (even up to an invertible transformation) was highly sensitive to model parameters.

We speculate that there are a few reasons for this. First, for the particular model being considered, there is a narrow parameter range where both (i) the components of $\boldsymbol{W}$ are small, and (ii) the PPC constraints are satisfied. For example, increasing the tuning curve width makes $\boldsymbol{W}$ smaller, but thwarts the first condition in Equation (3). Second, because moving from the $N \times N$ covariance matrix product to the $k \times k$ metric is a highly lossy operation, small errors can greatly affect the result. Although our proposed metric recovery approach is promising, improving its practical performance is the subject of ongoing work.

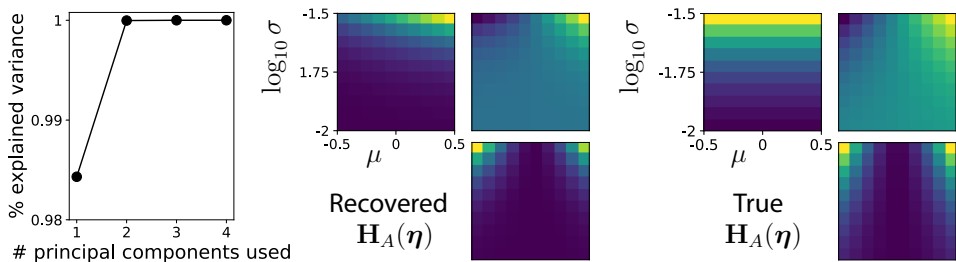

Figure 3: Example metric recovery. Left: variance explained by retaining $k = 2$ components of the adjusted signal covariance matrix. Middle: the recovered metric as a function of true $\mu$ and $\log_{10} \sigma$ (clockwise: $H_{11}$, $H_{12}$, $H_{22}$). Right: true metric.

## 7. Discussion

We identified a natural candidate for the neural manifold of a PPC—the statistical manifold of the represented probability distribution—and showed that it is, at least in principle, possible to recover its geometry from neural data by measuring tuning curves and covariance matrices. Interestingly, one step of the proposed recovery method is fairly similar to principal component analysis, except that it involves the eigendecomposition of the 'denoised' signal covariance matrix rather than the signal covariance matrix itself. This denoising is unnecessary when noise correlations are negligible (i.e. when spiking statistics are well-described by independent Poisson neurons), but could be important for learning latent geometry when there is a nontrivial noise correlation structure.

There remain many open questions, some of which suggest clear directions for future work. Most importantly: can the typical performance of our proposed recovery approach be substantially improved, so that the answer to this paper's title is a clear 'yes' instead of a 'maybe'? Robustness and stability might be improved by augmenting this approach with a more typical one, e.g. maximum likelihood recovery of the represented distribution (Walker et al., 2020). It also remains to be seen whether a method like this could be successfully applied to real neural data, which features a variety of additional complications.

An orthogonal direction for future study is improving and generalizing the theoretical result we obtained, which relates information geometry to a specific kind of neural population code (PPCs) in a specific parameter regime (small readout weights). It may be possible to derive analogous results for population codes which represent parametric probability distributions in other ways, including distributed distributional codes (Zemel et al., 1998; Vértes and Sahani, 2018), quantile codes, and expectile codes (Dabney et al., 2020; Lowet et al., 2020). It may also be possible to say something interesting about the case where the PPC readout weights are not small, and the geometry of the latent neural manifold is no longer as closely related to the denoised signal covariance matrix (Equation (13)). On the other hand, it may no longer even make sense to identify the latent neural manifold with the statistical manifold in such a regime; we leave such questions, which are both philosophical and technical in nature, for future work.

### Acknowledgments

This work was supported by grants from the National Institutes of Health (JD: 1U19NS118246; ZC: 5T32MH020017, 5T32EY007110).

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

## Appendix A. Exponential family example: normal distribution

In this appendix, we present a familiar distribution—the normal distribution—in terms of exponential family concepts. The likelihood associated with a normally distributed random variable $s \in \mathbb{R}$ can be written in exponential family form as

$$p(s|\boldsymbol{\eta}) = \frac{1}{\sqrt{2\pi\sigma^2}} \exp\left\{-\frac{(s-\mu)^2}{2\sigma^2}\right\} = h \exp\left\{\boldsymbol{\eta}^T \boldsymbol{T}(s) - A(\boldsymbol{\eta})\right\} \tag{24}$$

where

$$\boldsymbol{T}(s) := (s, s^2)^T \tag{25}$$

$$\boldsymbol{\eta} := (\eta_1, \eta_2)^T = \left(\frac{\mu}{\sigma^2}, -\frac{1}{2\sigma^2}\right)^T$$

$$A(\boldsymbol{\eta}) := \frac{\mu^2}{2\sigma^2} + \log\sigma = -\frac{\eta_1^2}{4\eta_2} - \frac{1}{2}\log(-2\eta_2)$$

$$h = \frac{1}{\sqrt{2\pi}} \ .$$

This is the 'canonical' choice of $\boldsymbol{T}(s)$, $\boldsymbol{\eta}(s)$, and so on, but there is an entire equivalence class of choices that yield the same expression for $p(s|\boldsymbol{\eta})$. For example, multiplying $\boldsymbol{T}(s)$ by 2 and dividing $\boldsymbol{\eta}$ by 2 would yield the same distribution.

The Hessian of the log-partition function (i.e. the Fisher information) with respect to the natural parameters is

$$\boldsymbol{\mathcal{I}}(\boldsymbol{\eta}) = \boldsymbol{H}_A(\boldsymbol{\eta}) = \begin{pmatrix} -\frac{1}{2\eta_2} & \frac{\eta_1}{2\eta_2^2} \\ \frac{\eta_1}{2\eta_2^2} & \frac{1}{2\eta_2^2}\left(1 - \frac{\eta_1^2}{\eta_2}\right) \end{pmatrix} = \begin{pmatrix} \sigma^2 & 2\mu\sigma^2 \\ 2\mu\sigma^2 & 2\sigma^4\left(1 + \frac{2\mu^2}{\sigma^2}\right) \end{pmatrix} \ . \tag{26}$$

## Appendix B. PPC example: representing a normal distribution

In this appendix, we present one of the simplest nontrivial PPCs, which is a population of independent Poisson neurons that encodes a normal distribution with mean $\mu$ and variance $\sigma^2$. Consider $N$ independent Poisson neurons with tuning curves

$$\langle r_i \rangle_{p(\boldsymbol{r}|s)} = f_i(s) = \frac{g}{\sqrt{2\pi a^2/(\Delta x)}} \exp\left\{-\frac{(s-x_i)^2}{2a^2}\Delta x\right\} \tag{27}$$

where $x_i := x_{min} + (x_{max} - x_{min})(i/N)$ is the center of the $i$-th neuron's tuning curve, $a/\sqrt{\Delta x}$ is the width of each tuning curve, and $g > 0$ is a gain parameter. The number $\Delta x$ is chosen so that

$$\mathbf{1}^T \boldsymbol{f}(s)\Delta x = \sum_{i=1}^{N} f_i(s)\Delta x = g \tag{28}$$

i.e. $\Delta x = (x_{max} - x_{min})/N$. (For this to work, $N$ must be sufficiently large, and $x_{min}$ and $x_{max}$ must be chosen so that $x_{min}$ is somewhat smaller than the smallest value of $s$, and $x_{max}$ is somewhat larger than the largest value of $s$.) This makes $g/\Delta x$ the expected

total number of spikes in the population. Independent Poisson neurons encode a normal distribution, since

$$p(s|\boldsymbol{r}) \propto_s \prod_i p(r_i|s) \propto_s \exp\left\{ \left( \frac{\boldsymbol{x}^T \boldsymbol{r}}{a^2} \Delta x \right) s + \left( -\frac{\mathbf{1}^T \boldsymbol{r}}{2a^2} \Delta x \right) s^2 \right\} \ . \tag{29}$$

The mean and variance of this distribution are

$$\mu = \frac{\boldsymbol{x}^T \boldsymbol{r}}{\mathbf{1}^T \boldsymbol{r}} \tag{30}$$

$$\sigma^2 = \frac{a^2}{\mathbf{1}^T \boldsymbol{r} \ \Delta x} \ .$$

The corresponding natural parameters are

$$\eta_1 = \frac{\boldsymbol{x}^T \boldsymbol{r}}{a^2} \Delta x \tag{31}$$

$$\eta_2 = -\frac{\mathbf{1}^T \boldsymbol{r}}{2a^2} \Delta x \ .$$

Since the parameter estimates satisfy $\hat{\boldsymbol{\eta}} = \boldsymbol{W}\boldsymbol{r}$, the rows of the readout matrix $\boldsymbol{W}$ are

$$\boldsymbol{w}_1 = \frac{1}{a^2} \ \boldsymbol{x} \ \Delta x \tag{32}$$

$$\boldsymbol{w}_2 = -\frac{1}{2a^2} \ \mathbf{1} \ \Delta x \ .$$

## Appendix C. Deriving formal mean and covariance expressions

The linear PPC condition on $p(\boldsymbol{s}|\boldsymbol{r})$ (Equation (2)) is equivalent, at least for sufficiently well-behaved exponential families where $\boldsymbol{s}$ is continuous, to

$$\nabla_{\boldsymbol{s}} p(\boldsymbol{s}|\boldsymbol{r}) = \boldsymbol{J}_{\boldsymbol{T}}^T(\boldsymbol{s}) \ \boldsymbol{W}\boldsymbol{r} \ p(\boldsymbol{s}|\boldsymbol{r}) \tag{33}$$

where $\boldsymbol{J}_{\boldsymbol{T}}(\boldsymbol{s})$ is the $k \times S$ Jacobian of $\boldsymbol{T}(\boldsymbol{s})$. For a uniform stimulus prior $p(\boldsymbol{s})$, Bayes' rule indicates that the above is equivalent to

$$\nabla_{\boldsymbol{s}} p(\boldsymbol{r}|\boldsymbol{s}) = \boldsymbol{J}_{\boldsymbol{T}}^T(\boldsymbol{s}) \ \boldsymbol{W}\boldsymbol{r} \ p(\boldsymbol{r}|\boldsymbol{s}) \ . \tag{34}$$

The above equation can be viewed as a constraint on the types of neural activity compatible with the desired $p(\boldsymbol{s}|\boldsymbol{r})$. It implies necessary (but not sufficient) constraints on tuning curves and covariance matrices, among other things (e.g. Equation (3)).

It is helpful to rewrite Equation (34) in terms of the corresponding (factorial-cumulant) generating function, which is a general-purpose tool for studying a wide variety of biophysically-relevant stochastic processes (see e.g. Singh and Bokes (2012); Gorin et al. (2021)). Define the probability-generating function

$$\psi(\boldsymbol{u}, \boldsymbol{s}) := \sum_{\boldsymbol{r}} (\boldsymbol{u} + \mathbf{1})^{\boldsymbol{r}} p(\boldsymbol{r}|\boldsymbol{s}) = \sum_{r_1,\dots,r_N} (u_1 + 1)^{r_1} \cdots (u_N + 1)^{r_N} p(\boldsymbol{r}|\boldsymbol{s}) \tag{35}$$

This always exists for $\boldsymbol{u}+\boldsymbol{1}$ chosen to be on the complex unit sphere (the subset of $\mathbb{C}^N$ with norm 1). Define the factorial-cumulant generating function via $\phi(\boldsymbol{u},\boldsymbol{s}) := \log\psi(\boldsymbol{u},\boldsymbol{s})$. This object is useful to define since derivatives (with respect to $\boldsymbol{u}$) of $\phi$ correspond to special moments of $p(\boldsymbol{r}|\boldsymbol{s})$. In particular,

$$f_i(\boldsymbol{s}) := \langle r_i \rangle_{p(\boldsymbol{r}|\boldsymbol{s})} = \left.\frac{\partial\phi}{\partial u_i}\right|_{\boldsymbol{u}=\boldsymbol{0}} \tag{36}$$

$$\Sigma_{ij}(\boldsymbol{s}) := \mathrm{Cov}(r_i,r_j)_{p(\boldsymbol{r}|\boldsymbol{s})} - \delta_{ij}\langle r_i\rangle_{p(\boldsymbol{r}|\boldsymbol{s})} = \left.\frac{\partial^2\phi}{\partial u_i\partial u_j}\right|_{\boldsymbol{u}=\boldsymbol{0}} . \tag{37}$$

Rewriting Equation (34) in terms of $\phi$ yields

$$\nabla_{\boldsymbol{s}}\phi(\boldsymbol{u},\boldsymbol{s}) = \boldsymbol{J}_{\boldsymbol{T}}^T(\boldsymbol{s})\,\boldsymbol{W}\,[\,(\boldsymbol{u}+\boldsymbol{1})\odot\nabla_{\boldsymbol{u}}\phi(\boldsymbol{u},\boldsymbol{s})\,] \tag{38}$$

where $\odot$ denotes the element-wise/Hadamard product. It is important to note that $\phi$ only depends on certain combinations of $\boldsymbol{s}$ and $\boldsymbol{u}$. In particular, define the variables

$$\nu_j := (u_j+1)\exp\left\{\sum_i W_{ij}T_i(\boldsymbol{s})\right\} \tag{39}$$

for all $j = 1,...,N$. If $\phi$ only depends on $\boldsymbol{u}$ and $\boldsymbol{s}$ through the $\nu_j$, then Equation (38) is solved, since

$$\nabla_{\boldsymbol{s}}\phi = \sum_j \frac{\partial\phi}{\partial\nu_j}\nabla_{\boldsymbol{s}}\nu_j = \boldsymbol{J}_{\boldsymbol{T}}^T(\boldsymbol{s})\,\boldsymbol{W}\,[\,(\boldsymbol{u}+\boldsymbol{1})\odot\nabla_{\boldsymbol{u}}\phi\,] . \tag{40}$$

Conversely, if $\phi$ satisfies Equation (38), then it can be written as a function of the $\nu_j$ only, since

$$d\phi = \sum_i \frac{\partial\phi}{\partial s_i}ds_i + \sum_j \frac{\partial\phi}{\partial u_j}du_j \tag{41}$$

$$= \sum_i \left[\sum_{m,j} J_{im}^T W_{mj}(u_j+1)\frac{\partial\phi}{\partial u_j}\right]ds_i + \sum_j \frac{\partial\phi}{\partial u_j}du_j$$

$$= \sum_j \left\{\left[\sum_{i,m} J_{im}^T W_{mj}ds_i\right](u_j+1) + du_j\right\}\frac{\partial\phi}{\partial u_j}$$

$$= \sum_j \exp\left\{\sum_m W_{mj}T_m(\boldsymbol{s})\right\}\frac{\partial\phi}{\partial u_j}d\nu_j$$

$$= \sum_j \frac{\partial\phi}{\partial\nu_j}d\nu_j .$$

Hence, the general solution of Equation (38) is given by some function $\phi = \phi(\nu_1,...,\nu_N)$. Because $\phi$ is analytic in $\boldsymbol{u}$ near $\boldsymbol{u}=\boldsymbol{0}$ for all $\boldsymbol{s}$, it is also analytic in $\boldsymbol{\nu}$ in a neighborhood of $\boldsymbol{u}=\boldsymbol{0}$, and can be formally written as the Taylor expansion

$$\phi(\boldsymbol{\nu}) = \sum_{\boldsymbol{n}\in\mathbb{N}^N} c_{\boldsymbol{n}}\,\boldsymbol{\nu}^{\boldsymbol{n}} = \sum_{\boldsymbol{n}\in\mathbb{N}^N} c_{\boldsymbol{n}}\,\nu_1^{n_1}\cdots\nu_N^{n_N} \tag{42}$$

for some coefficients $c_{\boldsymbol{n}}$. In terms of $\boldsymbol{u}$ and $\boldsymbol{s}$, we have

$$\phi(\boldsymbol{u}, \boldsymbol{s}) = \sum_{\boldsymbol{n} \in \mathbb{N}^N} c_{\boldsymbol{n}} \, e^{\boldsymbol{n}^T \boldsymbol{W}^T \boldsymbol{T}(\boldsymbol{s})} \left[(\boldsymbol{u} + \boldsymbol{1})^{\boldsymbol{n}} - 1\right] \tag{43}$$

where the minus one is included to account for the constraint that (due to probabilities summing to one) $\phi(\boldsymbol{u} = 0, \boldsymbol{s}) = 0$. By taking the appropriate derivatives with respect to $\boldsymbol{u}$ (see Equation (36)), we obtain the moment results used in the main text (Equation (10)).

## Appendix D. The readout matrix can be chosen to be semi-orthogonal

The $k \times N$ readout matrix $\boldsymbol{W}$ is only defined up to an invertible $k \times k$ matrix $\boldsymbol{R}$. In this appendix, we will show that if $\boldsymbol{W}$ is one possible readout matrix, then there exists an invertible linear transformation $\boldsymbol{R}$ such that $\boldsymbol{O} := \boldsymbol{R}\boldsymbol{W}$ is semi-orthogonal, i.e. $\boldsymbol{O}\boldsymbol{O}^T = \boldsymbol{I}_k$.

First, note that $\boldsymbol{W}$ must have rank $k$; otherwise, the space of natural parameters $\hat{\boldsymbol{\eta}} := \boldsymbol{W}\boldsymbol{r}$ would have dimension less than $k$, in which case the represented distribution would have less than $k$ sufficient statistics.

Let $\boldsymbol{M}$ be any $k \times k$ positive-definite matrix, which necessarily has rank $k$. Since the rank of both $\boldsymbol{W}$ and $\boldsymbol{M}$ are $k$, the product $\boldsymbol{W}^T \boldsymbol{M} \boldsymbol{W}$ has rank $k$. This means that the product has a compact singular value decomposition

$$\boldsymbol{W}^T \boldsymbol{M} \boldsymbol{W} = \boldsymbol{U}^T \boldsymbol{D} \boldsymbol{V} \tag{44}$$

where $\boldsymbol{D}$ is a $k \times k$ diagonal matrix with only nonzero values on its diagonal, and $\boldsymbol{U}$ and $\boldsymbol{V}$ are both $k \times N$ semi-orthogonal matrices. By exploiting the semi-orthogonality of $\boldsymbol{U}$ and the invertibility of $\boldsymbol{D}$, the above expression can be rewritten as

$$\left(\boldsymbol{D}^{-1} \boldsymbol{U} \boldsymbol{W}^T \boldsymbol{M}\right) \boldsymbol{W} = \boldsymbol{V} \ . \tag{45}$$

Our claim is true if the $k \times k$ matrix in parentheses is invertible. But the matrix product is indeed invertible, since each factor is a full-rank matrix, so that the overall matrix has rank $k$. Hence, there always exists an invertible $k \times k$ matrix that makes $\boldsymbol{W}$ semi-orthogonal. Moreover, $\boldsymbol{W}$ can specifically be chosen to be one of the semi-orthogonal matrices that appears in the compact singular value decomposition of $\boldsymbol{W}^T \boldsymbol{M} \boldsymbol{W}$.

## Appendix E. A novel PPC with a nontrivial neural correlation structure

In this appendix, we describe a novel PPC—the *minimal correlation model*—that we use to illustrate our metric recovery strategy. What makes this model useful from the point of view of studying manifold recovery is that it is (i) analytically tractable, and (ii) has a nontrivial neural correlation structure.

The idea is the following. The case of independent Poisson neurons (as in e.g. Appendix B) is well-known and analytically tractable, but too trivial to be realistic. Can we come up with a minimal extension of it which exhibits a nontrivial correlation structure, but remains tractable?

One fact about the independent Poisson model is that its factorial-cumulant generating function (as defined in Appendix C) is

$$\phi_{ind}(\boldsymbol{u}, \boldsymbol{s}) = \boldsymbol{f}(\boldsymbol{s})^T \boldsymbol{u} \ . \tag{46}$$

Importantly, it is linear in $\boldsymbol{u}$. This means that all higher order factorial-cumulants (e.g. the variance of $r_1$ minus the mean of $r_1$) are zero. A reasonable extension of the above model is to one with a general second-order term:

$$\phi(\boldsymbol{u}, \boldsymbol{s}) = \boldsymbol{f}(\boldsymbol{s})^T \boldsymbol{u} + \frac{1}{2} \boldsymbol{u}^T \left( \boldsymbol{\Sigma}(\boldsymbol{s}) - \mathrm{diag}(\boldsymbol{f}(\boldsymbol{s})) \right) \boldsymbol{u} = \boldsymbol{f}(\boldsymbol{s})^T \boldsymbol{u} + \frac{1}{2} \boldsymbol{u}^T \tilde{\boldsymbol{\Sigma}}(\boldsymbol{s}) \boldsymbol{u} \qquad (47)$$

where

$$\tilde{\boldsymbol{\Sigma}}(\boldsymbol{s}) := \boldsymbol{\Sigma}(\boldsymbol{s}) - \mathrm{diag}(\boldsymbol{f}(\boldsymbol{s})) \qquad (48)$$

could be called the *excess covariance matrix*. It can be shown (see Equation (36)), by taking two derivatives with respect to $\boldsymbol{u}$, that the covariance matrix of the above model is $\boldsymbol{\Sigma}(\boldsymbol{s})$. Hence, since this model (before enforcing PPC-related constraints) permits an arbitrary mean and covariance structure, it is a sort of discrete analogue of the multivariate normal distribution.

Suppose we would like a neural population whose statistics are described by Equation (48) to represent a normal distribution with natural parameters

$$\eta_1 = \frac{\boldsymbol{x}^T \boldsymbol{r}}{a^2} \Delta x \qquad (49)$$

$$\eta_2 = -\frac{\mathbf{1}^T \boldsymbol{r}}{2a^2} \Delta x \ .$$

We will choose the desired readout matrix $\boldsymbol{W} = \begin{pmatrix} \boldsymbol{w}_1^T \\ \boldsymbol{w}_2^T \end{pmatrix}$ and vector of sufficient statistics $\boldsymbol{T}(s)$ to be (as in Appendix B)

$$\boldsymbol{w}_1 = \frac{1}{a^2} \ \boldsymbol{x} \ \Delta x \qquad (50)$$

$$\boldsymbol{w}_2 = -\frac{1}{2a^2} \ \mathbf{1} \ \Delta x$$

$$\boldsymbol{T}(s) = (s, s^2)^T \ .$$

For our population to encode a normal distribution in the sense just described, it is necessary and sufficient for Equation (48) to satisfy the generating function version of the PPC condition (Equation (38)). Equivalently, this equation puts necessary and sufficient constraints on the tuning curves $\boldsymbol{f}(s)$ and covariance structure $\boldsymbol{\Sigma}(s)$. In particular, the constraints are that

$$0 = \frac{\partial \boldsymbol{v}(s)^T}{\partial s} \ \boldsymbol{f}(s) \qquad (51)$$

$$\frac{\partial \boldsymbol{f}(s)}{\partial s} = \boldsymbol{\Sigma}(s) \ \frac{\partial \boldsymbol{v}(s)}{\partial s} \qquad (52)$$

$$\frac{\partial \tilde{\boldsymbol{\Sigma}}(s)}{\partial s} = \mathrm{diag}\left( \frac{\partial \boldsymbol{v}(s)}{\partial s} \right) \ \tilde{\boldsymbol{\Sigma}}(s) + \tilde{\boldsymbol{\Sigma}}(s) \ \mathrm{diag}\left( \frac{\partial \boldsymbol{v}(s)}{\partial s} \right) \qquad (53)$$

where we have defined

$$\boldsymbol{v}(s) := \boldsymbol{W}^T \boldsymbol{T}(s) \ . \qquad (54)$$

One possible solution of the above set of equations, which can be verified by tedious but straightforward algebra, is

$$f_i(s) = \frac{g}{\sqrt{2\pi a^2/\Delta x}} \left\{ (1-c) \exp\left\{ -\frac{(x_i - s)^2}{2a^2/\Delta x} \right\} + \frac{c}{\sqrt{\frac{1+d}{2}}} \exp\left\{ -\frac{(x_i - s)^2}{2\frac{a^2}{\Delta x}\frac{(1+d)}{2}} \right\} \right\}$$
(55)

$$\tilde{\Sigma}_{ij}(s) = g\frac{c}{\sqrt{(2\pi a^2/\Delta x)^2 d}} \exp\left\{ -\frac{\left(\frac{x_i + x_j}{2} - s\right)^2}{a^2/\Delta x} - \frac{(x_i - x_j)^2}{4a^2d/\Delta x} \right\} .$$
(56)

There are two new parameters, $c \in [0, 1]$ and $d > 0$, which do not appear in the independent Poisson model described in Appendix B. The parameter $c$ could be considered to define the overall 'strength' of correlations. If it is zero, the model reduces to the independent Poisson model; as it is increased, the off-diagonal elements of the covariance matrix become larger.

The parameter $d$ determines how off-diagonal the structure of the excess covariance matrix is. This is clear since $\tilde{\Sigma}_{ij}(s)$ is a product of two Gaussians: the first takes its maximum value where $(x_i + x_j)/2 = s$ and has variance $a^2/(2\Delta x)$; the second takes its maximum value where $x_i = x_j$ and has variance $2a^2d/\Delta x$. When $d$ is small, the first Gaussian dominates, and $\tilde{\Sigma}$ is 'hottest' around the diagonal. When $d$ is large, the second Gaussian dominates, and $\tilde{\Sigma}$ instead takes its largest values around the anti-diagonal (see Figure 2).

These expressions can be used to analytically compute that

$$\langle f_i(s) \rangle_{p(s|\eta)} = g\left\{ \frac{(1-c)}{\sqrt{2\pi\left(\sigma^2 + \frac{a^2}{\Delta x}\right)}} \exp\left\{ -\frac{(x_i - \mu)^2}{2\left(\sigma^2 + \frac{a^2}{\Delta x}\right)} \right\} \right.$$
(57)

$$\left. + \frac{c}{\sqrt{2\pi\left(\sigma^2 + \frac{a^2}{\Delta x}\frac{(1+d)}{2}\right)}} \exp\left\{ -\frac{(x_i - \mu)^2}{2\left(\sigma^2 + \frac{a^2}{\Delta x}\frac{(1+d)}{2}\right)} \right\} \right\}$$

$$[\boldsymbol{\Sigma}_{noise}(\boldsymbol{\eta})]_{i,j} = g\frac{c}{\sqrt{(2\pi)^2\left(\sigma^2 + \frac{a^2}{2\Delta x}\right)2\frac{a^2d}{\Delta x}}} \exp\left\{ -\frac{\left(\frac{x_i + x_j}{2} - \mu\right)^2}{2\left(\sigma^2 + \frac{a^2}{2\Delta x}\right)} - \frac{(x_i - x_j)^2}{4a^2d/\Delta x} \right\}$$
(58)

$$+ \delta_{ij} \langle f_i(s) \rangle_{p(s|\boldsymbol{\eta})} .$$

They can also be used to numerically compute $\boldsymbol{\Sigma}_{signal}(\boldsymbol{\eta})$, as in Figure 2.

As a final detail, in order for this population to encode a normal distribution with a specific mean $\mu$ and a specific variance $\sigma^2$, it must be the case that

$$\langle \boldsymbol{w}_1^T \boldsymbol{r} \rangle_{p(\boldsymbol{r}|\boldsymbol{\eta})} = \frac{\mu}{\sigma^2}$$
(59)

$$\langle \boldsymbol{w}_2^T \boldsymbol{r} \rangle_{p(\boldsymbol{r}|\boldsymbol{\eta})} = -\frac{1}{2\sigma^2} .$$

Using Equation (57), this means that

$$\frac{g\mu}{a^2} = \frac{\mu}{\sigma^2}$$

$$-\frac{g}{2a^2} = -\frac{1}{2\sigma^2} \ .$$

(60)

Hence, it is sufficient to choose $g$ such that $g = (a/\sigma)^2$.

