# OpenReview forum: "Is the information geometry of probabilistic population codes learnable?"
_NeurIPS.cc/2022/Workshop/NeurReps — NeurReps 2022 Oral_

### Official Review · Reviewer_ZZfU · 2022-10-06
**Can we learn the Fisher metric of a probability population code modeled as an exponential family distribution?**

**Confidence:** 4
**Soundness:** 4
**Presentation:** 4
**Contribution:** 3
**Overall Rating:** 8

**Summary:**

This paper deals with learning the latent neural manifold from the observations of a large population of neurons (neural activity) modeled as a probabilisitic population code (PPC).
Key assumption is (3)
The Fisher metric is the Hessian of a log partition of an exponential family. Ambiguity of the sufficient statistics with the natural parameter via an invertible matrix is mentioned and a way to circumvent it is presented.
Experiments is detailed in Sec. 6 with self criticism mentioned in page 9.


**Questions:**

- Can you consider estimating semi-parametric distributions with  nuisance parameters?
See
Begun, Janet M., et al. "Information and asymptotic efficiency in parametric-nonparametric models." The Annals of Statistics 11.2 (1983): 432-452.

- Can you consider the renormalization group flow to study more coarse-grained geometry?
See
Machta, Benjamin B., et al. "Parameter space compression underlies emergent theories and predictive models." Science 342.6158 (2013): 604-607.



What is called information geometry is the Fisher-Rao manifold here (statistical manifold in information geometry has a different meaning, See Lauritzen definition)
In fig. 1b it is squared distance
Eq 9 is moment generating function of an exponential family

**Limitations:**

- What are other population models besides Linear PPCs?

**Recommended Decision:**

3: Accept

**Relevance:**

4: Highly relevant

**Strengths And Weaknesses:**

Strength:
- very well written
- important problem well-suited for the workshop at the intersection of systems neuroscience and geometry
- Algorithm 1 and experiments in Section 6 with discussion on limitations

Weakness:
- Lack a conclusion to the question asked in the title. Got the feeling that is is currently "maybe" and that future work would improve to "yes" under conditions..
- Other models than linear PPCs?

**Submission Track:**

Proceedings Paper (9 Page)

---

### Official Review · Reviewer_R4sW · 2022-10-14
**Is the information geometry of probabilistic population codes learnable?**

**Confidence:** 1
**Soundness:** 3
**Presentation:** 4
**Contribution:** 3
**Overall Rating:** 7

**Summary:**

The author(s) present a method for learning a metric on R^k, representing this space as a statistical manifold associated a probability distribution. This probability distribution is a "Probabilistic population code" driving measured neural activity.

**Questions:**

I would like the authors to state more clearly what is novel about their contribution, and to whom it is relevant. Specifically, is there anything here which could be of use to a non-neuroscientist? Are the authors confident that this analysis hasn't already been done in a more general setting?

**Limitations:**

The authors are very clear about the limitations of their work; in particular, there is absolutely no doubt which steps are approximations.

**Recommended Decision:**

3: Accept

**Relevance:**

2: Limited relevance

**Strengths And Weaknesses:**

Strengths:
1. The paper is very well-written
2. The idea and methodology is exciting.

Weaknesses:
1. The authors are very open about this, but there is no estimation theory (e.g. statistical consistency); I think this is ok.
2. It wasn't obvious to me why the paper was so centered on the neuroscience application - the problem treated seems more general.
3. Related to the above, I don't feel sure that the same problem wouldn't have been studied in other statistical contexts. (This is the most important, potential weakness.)
4. Still related to the above, some of the terminology and notation is a little opaque to me - e.g. <> for mean, or the word "only defined up to" rather than "identifiable only up to" (if I understand correctly)


**Submission Track:**

Proceedings Paper (9 Page)

---

### Official Review · Reviewer_qPk5 · 2022-10-16
**A valuable contribution towards learning manifold geometry from neural data**

**Confidence:** 4
**Soundness:** 4
**Presentation:** 4
**Contribution:** 3
**Overall Rating:** 8

**Summary:**

Adopting the view that populations of neurons encode information in the form of probabilistic population codes (PPCs), the authors identify the latent neural manifold of a neural population with the statistical manifold, endowed with the Fisher information metric, of the represented distribution. The authors analytically demonstrate for the case of linear PPCs how the metric of the neural manifold can be recovered from covariance matrices of neural activity and the readout matrix.

Crucially relying on the assumption that the components of the readout matrix W are small, the authors introduce an approximation that allows them to express the natural metric on the neural manifold to the signal covariance matrix for linear PPCs, which involves a procedure to “undo” the contamination by noise correlations on signal correlations, and a PCA-like approach to obtain the readout matrix. This allows, in principle, for a characterization of the neural manifold geometry using neural activity data. The authors introduce a synthetic dataset of a neural population, with tractable but nontrivial correlation structure, that encodes a normal distribution, and run experiments to illustrate the proposed metric-learning method.

**Questions:**

Under Eq. 20, it is stated: “this choice corresponds to choosing the coordinate system of the latent space so that the neural manifold is Euclidean on average.”

My impression is that this is a strong limitation. Under what conditions are you free to make this choice? Does such a coordinate system always exist?

**Limitations:**

The authors adequately address the limitations of the presented work, with respect to the experimental performance. I believe some more discussion on the assumption that the readout matrix is small would be illuminating.

I would further suggest the authors define some kind of performance metric for the recovered neural geometry, i.e., quantitative description of the deviation from the true geometry.

**Recommended Decision:**

3: Accept

**Relevance:**

4: Highly relevant

**Strengths And Weaknesses:**

This paper makes a strong theoretical contribution to this area of research. The paper is very well written, paints a clear narrative, and constructs a persuasive analytical argument for the discovery of latent neural manifold geometry from neural data. In my view, it presents a valuable novel framework from which to extract geometric insights of neural representations.
A weakness of this paper is the lack of convincing experimental validation of the proposed method. The method fails to recover the true representational geometry in many cases, even for the simplest non-trivial neural correlation structure. Nevertheless, I believe the method is promising and opens the door to a very interesting line of research.

**Submission Track:**

Proceedings Paper (9 Page)

---

### Decision · Program_Chairs · 2022-10-21

Accept (Oral)